# Designing host-associated microbiomes using the consumer/resource model

Germán Plata,[1] Karthik Srinivasan,[2] Madan Krishnamurthy,[3] Lukas Herron,[4] Purushottam Dixit[2,5]

**ABSTRACT** A key step toward rational microbiome engineering is *in silico* sampling of realistic microbial communities that correspond to desired host phenotypes, and vice versa. This remains challenging due to a lack of generative models that simultaneously capture compositions of host-associated microbiomes and host phenotypes. To that end, we present a generative model based on the mechanistic consumer/resource (C/R) framework. In the model, variation in microbial ecosystem composition arises due to differences in the availability of effective resources (inferred latent variables), while species' resource preferences remain conserved. Simultaneously, the latent variables are used to model phenotypic states of hosts. *In silico* microbiomes generated by our model accurately reproduce universal and dataset-specific statistics of bacterial communities. The model allows us to address three salient questions in host-associated microbial ecologies: (i) which host phenotypes maximally constrain the composition of the host-associated microbiomes? (ii) how context-specific are phenotype/microbiome associations, and (iii) what are plausible microbiome compositions that correspond to desired host phenotypes? Our approach aids the analysis and design of microbial communities associated with host phenotypes of interest.

**IMPORTANCE** Generative models are extremely popular in modern biology. They have been used to model the variation of protein sequences, entire genomes, and RNA sequencing profiles. Importantly, generative models have been used to extrapolate and interpolate to unobserved regimes of data to design biological systems with desired properties. For example, there has been a boom in machine-learning models aiding in the design of proteins with user-specified structures or functions. Host-associated microbiomes play important roles in animal health and disease, as well as the productivity and environmental footprint of livestock species. However, there are no generative models of host-associated microbiomes. One chief reason is that off-the-shelf machine-learning models are data hungry, and microbiome studies usually deal with large variability and small sample sizes. Moreover, microbiome compositions are heavily context dependent, with characteristics of the host and the abiotic environment leading to distinct patterns in host-microbiome associations. Consequently, off-the-shelf generative modeling has not been successfully applied to microbiomes. To address these challenges, we develop a generative model for host-associated microbiomes derived from the consumer/resource (C/R) framework. This derivation allows us to fit the model to readily available cross-sectional microbiome profile data. Using data from three animal hosts, we show that this mechanistic generative model has several salient features: the model identifies a latent space that represents variables that determine the growth and, therefore, relative abundances of microbial species. Probabilistic modeling of variation in this latent space allows us to generate realistic *in silico* microbial communities. The model can assign probabilities to microbiomes, thereby allowing us to discriminate between dissimilar ecosystems. Importantly, the model predictively captures host-asso-

**Peer Reviewer** Leo Lahti, University of Turku, Turku, Finland

Address correspondence to Purushottam Dixit, purushottam.dixit@yale.edu, or Germán Plata, german@biomedit.com.

Germán Plata and Karthik Srinivasan contributed equally to this article. Author order reflects the duration of involvement in the project.

G.P. is an employee and owns profit interest in BiomEdit LLC.

See the funding table on p. 18.

ciated microbiomes and the corresponding hosts' phenotypes, enabling the design of microbial communities associated with user-specified host characteristics.

**KEYWORDS** host-associated microbiomes, generative modeling, consumer/resource model

Host-associated microbiomes play important roles in host health and disease (1) and, for livestock species, feed efficiency and environmental footprint (2). Unfortunately, however, rational design of microbiomes to achieve desired host phenotypic states remains challenging due to extensive variability (3–5), high dimensionality and dynamic nature of microbiomes (4, 6, 7), co-evolution of hosts and microbiome (8, 9), and numerous microbe ↔ microbe (10) and host ↔ microbe interactions (8, 9, 11).

A key step in rational design of high-dimensional biological systems is generative modeling. Generative models are machine-learning models that probabilistically generate *in silico* data and reproduce key features of experimental observations (12). These models have been successfully used to model protein sequences (13–15), genomes (16), presence/absence patterns in microbial ecosystems (17), and single-cell transcription profiles (18). Generative models have been used for data augmentation and to identify functional covariation in the data [e.g., residue-residue contacts in proteins (19)]. Generative models can assign probabilities to unseen observations and, therefore, can be used, for example, to identify disease effect of genetic variants (20). Importantly, generative models have been used to design biological systems with desired properties, for example, proteins with higher enzymatic activity (13). Generative models can be extremely useful for microbiome engineering, for example, in identifying feasible microbiome community structures that correspond to desired community functions or host phenotypes or in identifying specific host phenotypes, environmental parameters, or microbial taxa that constrain community composition.

Off-the-shelf generative models based on generative adversarial neural networks (21–24), maximum entropy methods (25), and mixture Dirichlet distributions (26) have been used to study the composition of microbiomes. However, many of these models tend to have a very large number of parameters and run the risk of overparameterization when modeling microbiomes. This is because unlike other biological data (protein sequences, RNAseq, etc.), microbiome studies usually comprise significantly smaller sample sizes ($N \sim 100$) and exhibit extensive variation. Additionally, these generative models provide little insight into the environmental (27, 28) and host factors (29) that drive the diversity in microbiomes. Notably, state-of-the-art generative models in microbiome studies have not modeled the simultaneous variation in corresponding host phenotypes.

An alternative to data-driven generative modeling is mechanistic generative modeling. In these models, variation in microbiome composition is explained using variation in interpretable and mechanistic model parameters. The consumer/resource (C/R) framework is one of the most popular mechanistic models to understand variation in microbiome composition (28, 30–36). In C/R models, variation in microbiomes arises due to variation in resource availability, while the preferences of microbial species (consumers) toward these resources remain conserved across similar ecosystems. With an appropriate choice of parameters, ecosystem compositions generated using the C/R model reproduce many universal statistical features of real and synthetic microbiomes (28, 32, 36, 37) such as distributions of diversity metrics (28) and scaling relationships (27). Unfortunately, fitting the C/R model to data is difficult as it requires knowledge of high-resolution temporal dynamics of species abundances, or the concentrations of resources consumed by microbial species through time (38) or species-resource preferences (39). Consequently, training these models on specific microbiome data sets is difficult even for controlled *in vitro* communities (36). Therefore, studies using the C/R model cannot model covariation patterns observed in specific host-microbiome systems, instead randomly drawn parameters have been used to explain statistical features of the data (27, 28). Finally, the C/R framework is not able to model variation in host

phenotypes, except for the abundances of metabolites directly consumed/excreted by the microbiome.

In this work, we develop a generative model for host-associated microbiomes using the mechanistic C/R framework. To that end, we identify a latent space induced from the C/R model that encodes the temporal histories of effective resources affecting microbial growth. Variation along this latent space results in variation in microbiome compositions. Using data from three host-associated microbiomes, we show that probabilistic modeling of this latent space reproduces both universal statistical features and system-specific covariations between bacterial taxa. Using the model and the bovine rumen microbiome as a test case, we investigate three salient questions in microbiome design and engineering: (i) which host phenotypes maximally constrain the composition of the host-associated microbiome? (ii) how context-specific are phenotype/microbiome associations, and (iii) what are plausible microbiome compositions that correspond to desired host phenotypes?

To the best of our knowledge, this is the first generative model that can simultaneously model microbiome compositions and the environmental factors, for example, phenotypic states of the host. Therefore, we believe that our generative model will be an important tool for the design of host-associated microbiomes.

## RESULTS

### The C/R framework identifies a latent space to build a machine-learning model

We use the C/R framework to identify a latent space to model microbiome compositions (see "Model description and fit to data"; Fig. 1). In the C/R model, the change in abundance of an organism (growth rate) depends on the availability of resources $r_k$ and species preferences toward those resources $\theta_{ko}$. The resources themselves are depleted according to their consumption by all organisms and net flow. As discussed above, fitting the C/R model to data requires not only the knowledge of microbial abundances but also the identities and abundances of those resources, either across samples or over time. Typically, these measurements are not available, and, consequently, the C/R model is only used for its qualitative insights. Here, we show how to convert the C/R model to a latent variable model to describe species abundances. To do so, we integrate the equation describing the growth rate of a species $o$ in the C/R model and obtain an expression for the species abundance $n_o(t)$ at time $t$:

$$n_o(t) \approx \exp\left(\int_{-\infty}^{t} \mu_o(\tau)d\tau\right) = \exp\left(-\sum_{k=1}^{K} z_k\theta_{ko}\right) \tag{1}$$

where $\mu_o(\tau)$ is the time-dependent growth rate. We recognize

$$z_k = -\int_{-\infty}^{t} r_k(\tau)d\tau \tag{2}$$

as latent variables that represent the complex temporal dynamics of resources. We note that the latent variables, thus defined, are emergent variables; they correspond neither to the initial concentration nor to the steady-state concentrations of the resources. Instead, they represent the temporal history of resource fluctuations which, in turn, may depend on flow rates, resource sharing, and cross-feeding (31, 37).

Recognizing these latent variables allows us to circumvent the numerical difficulties in fitting the C/R model to microbiome data (39). Observe that the relative abundances of organisms $q_o$ are simple exponential functions of the latent variables:

$$q_o = \frac{1}{\Omega}\exp\left(-\sum_{k=1}^{K} z_k\theta_{ko}\right) \tag{3}$$

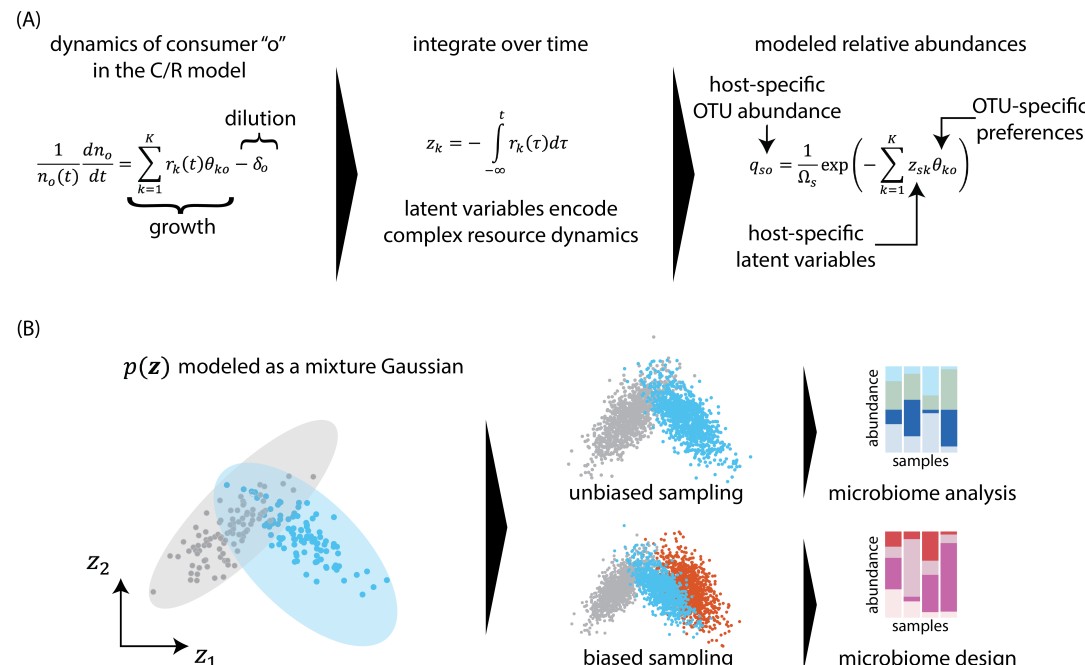

**FIG 1** A generative model for host-associated microbiomes using the C/R framework. The C/R model defines changes in the abundance $n_o(t)$ of consumers "o" as a function of resource abundances $r_k(t)$ and consumer preferences $\theta_{ko}$ toward resources. Integrating the C/R dynamics over time shows that species abundances are a function of latent variables $z_k(t)$ that represent the sample-specific time history of resources, and universal species-specific preferences toward those latent variables. (B) to generate realistic microbiomes, sample embeddings in the latent space are modeled using a mixture Gaussian distribution. The model allows data augmentation via unbiased sampling in the latent space. Biased sampling from the model allows us to generate microbiome samples that correspond to desired host phenotypes.

Unlike the C/R model, equation 3 can be easily fit to cross-sectional microbiome data (40). To that end, consider that the relative abundances (relative counts) $x_{so}$ of $O$ OTUs (operational taxonomic units) are measured in $S$ hosts. We assume that the resource preferences of OTUs are universal across hosts, while resource dynamics are host dependent. That is, latent variables $z_{sk}$ are host-specific, but species-independent and $\theta_{ko}$ are species-specific but host-independent. For fitting equation 3 with a fixed $K$, we minimize the cross-entropy $C$ (36), which is equivalent to maximizing a probabilistic model for species abundances that follow a multinomial distribution (equation 4).

$$C = \sum_{s,o} x_{so}\log q_{so}. \tag{4}$$

As we show in "Model description and fit to data," this minimization is a nonlinear low rank matrix factorization wherein $z$ and $\theta$ can be learned using gradient descent.

Unlike other machine-learning models, our model has clear mechanistic interpretation: microbial couplings to specific latent variables remain constant across different ecosystems, but the temporal histories of these latent variables vary from ecosystem-to-ecosystem, leading to observed variation in community compositions.

## Host-associated microbiomes are surprisingly low-dimensional

How accurate is equation 5 in modeling compositions of microbiomes? We studied microbiomes associated with three host species, the rumen of Holstein cows (41), the chicken cecum (42), and the human gut (43). The bovine and the chicken microbiomes were characterized at the level of operational taxonomic units (OTUs), while the human microbiomes were characterized at the genus level. Based on our previous analysis on technical noise associated with the sequencing of microbial samples, we only included

taxa whose average abundance was higher than 0.1% (5). In the three datasets, there were 790 samples and 157 OTUs (cows), 778 samples and 156 OTUs (chicken), and 127 samples and 95 genera (humans), respectively ("Details of the data sets").

The number of resources consumed by species in these ecosystems can potentially be very large (K ~ $10^3$–$10^4$) (36). To avoid overparameterization in our model, we seek a small number of effective resources (latent variables) (44) that capture the observed compositional variation of microbiomes. These latent variables do not correspond to specific nutrients present at a specific time. Instead, they represent a combination of factors that both positively and negatively affect the growth of microbial species aggregated over time. To determine the effective dimension of the latent space, we fit equation 5 where K is chosen as the smallest of the number of hosts S and the number of species O. We determined the effective dimension by examining the singular values (SVs) of $Z \times \Theta$ (insets of Fig. 2A through C). Singular values quantify the variance explained by low-rank approximations to matrices and, therefore, can identify coarse-grained latent variables that can accurately approximate the microbiome. SVs for all three ecosystems showed two regimes. When arranged from largest to the smallest, SVs first decreased rapidly, followed by a gradual power law-like decrease. This suggested that a small number of latent variables captured the essential variation in microbial composition. To confirm this, we fit microbiome compositions using different values of K. Indeed, as seen in Fig. 2D through F, the symmetric Kullback-Leibler divergence and the Bray-Curtis dissimilarity between data and model fits decreased approximately logarithmically with increasing dimension of the latent space and at a significantly slower rate for larger Ks.

These results show that a low-dimensional latent space constructed using the C/R model captures compositional variation in microbiomes in three different host species. We note that this dimensionality reduction approach is conceptually different from principal coordinate analysis (PCoA), a standard tool in microbiome analysis. In PCoA, one approximates dissimilarity between compositions of pairs of ecosystems using dimensionality reduction directly on the pairwise distance matrix. The variance

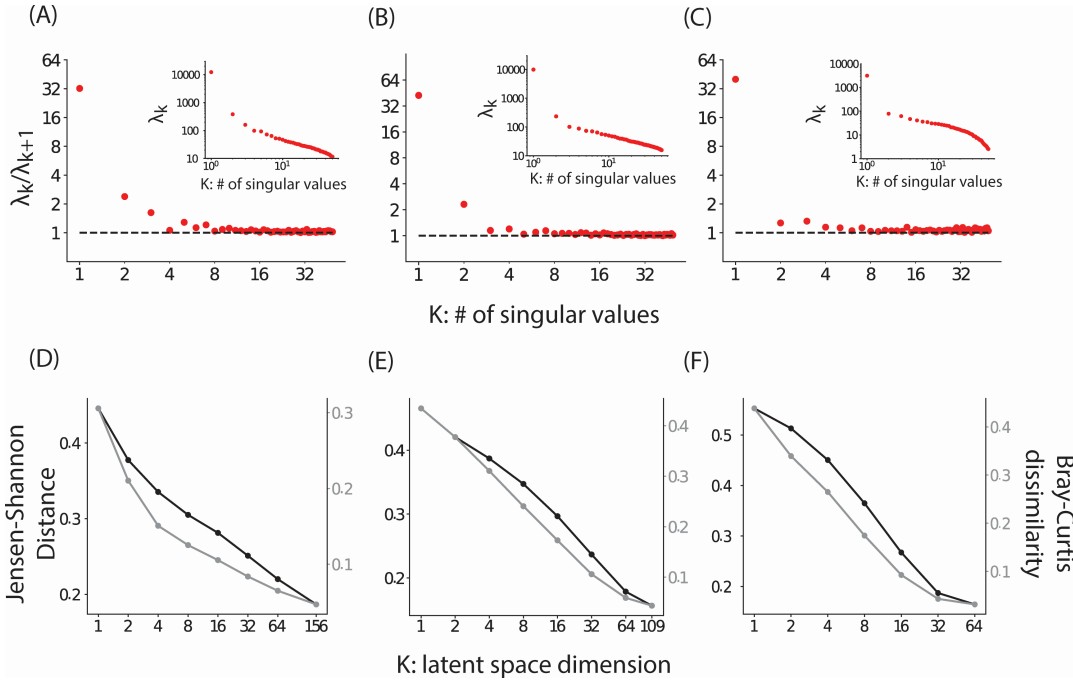

**FIG 2** Low dimensionality of host-associated microbiomes. (A to C) The ratio of successive singular values $\frac{\lambda_k}{\lambda_{k+1}}$ of the $Z \times \Theta$ matrix equation 3 for bovine, chicken, and human microbiomes, respectively. Only the first 50 SVs are shown as later, less important SVs may be dominated by noise. The insets show the singular values arranged in a decreasing order. (D to F) The average symmetric Jensen-Shannon distance (black) and the Bray-Curtis dissimilarity (gray) between observed community composition and the corresponding model fit as a function of the dimension of the latent space. The average is taken over all samples.

explained by PCoA is, therefore, the variance explained in predicting pairwise distances using a lower dimensional projection. In contrast, our approach directly models (approximates) compositions of individual ecosystems using a lower dimensional latent space. Notably, the accuracy of our approach quantified in terms of the variance explained in reconstructing community composition as well as reconstructing pairwise Bray-Curtis dissimilarity shows that a small number of components is sufficient to capture both community composition as well as $\beta$-diversity (Fig. S1).

This low dimensionality may arise due to covariation in species that results from specific structure in consumer/resource matrices (33, 35) due to covariation in resource flow rates (33), due to universal factors such as pH (45), or due to coarse-grained structure/function relationships between the microbiome and its environment (34) that affect broad classes of species in a similar manner. Notably, as we show below, this low dimensionality was not due to low compositional diversity. Indeed, species abundances were widely distributed with an abundance distribution (Fig. 3C) that had an extended power law scaling between abundances of 0.1% to 10%. In this work, we exploit this low dimensionality for generative modeling. We leave it for future studies to explore its mechanistic underpinnings.

Finally, we confirmed that the trends in community reconstruction accuracy did not depend on the abundance cutoff used as the OTU inclusion criteria (Fig. S2). Consequently, downstream analyses using these latent variables are also expected to be insensitive to the abundance cutoff used.

## Probabilistic modeling reproduces dataset-specific statistics of microbiomes

To endow generative capacity to our approach, we model the distribution $p(z)$ over latent variables using Gaussian mixture modeling (see "Mixture Gaussian models"). Notably, Gaussian mixtures are versatile models to fit multivariate data as they can in principle fit to distributions of arbitrary shapes. From this distribution, it is possible to generate realistic *in silico* microbiome compositions by first sampling their latent space embeddings using the learnt distribution and then using the universal species preferences learned during training and obtaining relative species abundances using equation 3. Additionally, the Gaussian mixture model also enables to calculate the likelihood of any given latent space coordinate profile relative to the data used to train the model.

Can the generative model sample realistic ecosystem compositions? We tested this using a comparison of several lower-order statistics, metrics of distribution density, and embedding accuracy. This is a common approach used in generative models. We trained a generative model using a latent space of dimension $K = 16$ (Fig. 2). We tested the accuracy of the generative model by sampling latent variables from an inferred mixture Gaussian $p(z)$ and then generating *in silico* microbiomes using equation 3. For clarity of presentation, we present results for chicken cecum microbiomes. The results for the other data sets are shown in Fig. S3 and S4. Figure 3A shows that the generative model accurately reproduces OTU mean abundances and standard deviations. Notably, our approach is quite accurate at predicting microbiome composition at all levels of taxonomy (Fig. S5). Figure 3B (and inset) shows that the model captures species abundance covariation and higher-order co-occurrence patterns. Figure 3C shows that the model reproduces the species abundance distribution, including the power law-like scaling between abundances of $10^{-3}$ and $10^{-1}$ with a power law exponent of $\gamma = -1.03$ ($\gamma \sim -1.12$ for the model) for the inverse cumulative distribution. Figure 3D shows that the model reproduces the distribution of the $\alpha$-diversity metric of Shannon diversity. Figure 3E shows that the model reproduces distributions of the $\beta$-diversity metric Bray-Curtis dissimilarity between nearest neighbor samples as well as pairs of randomly picked samples.

Are the species preferences toward effective resources $\theta_{ko}$ universal? If so, species preferences inferred from a set of microbiome samples should be able to accurately model the composition of dissimilar microbial communities of the same species. To test this hypothesis, we looked at bacterial microbiomes associated with the cecum and

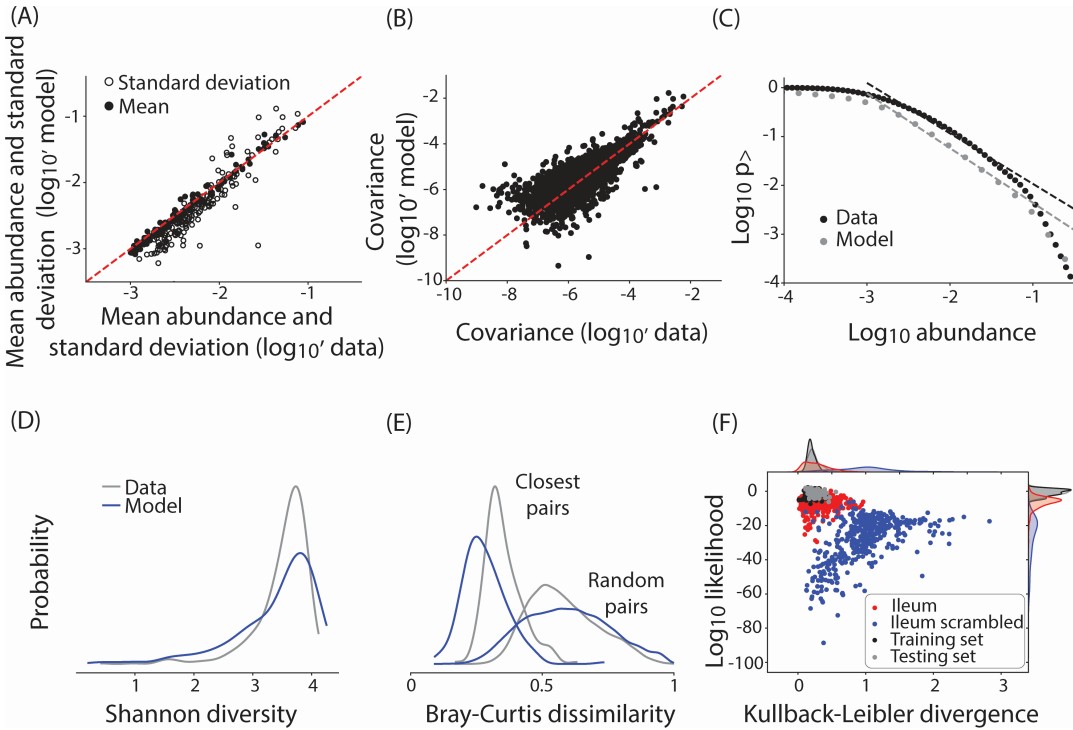

**FIG 3** Latent space-based model generates accurate *in silico* communities. (A) A comparison between mean OTU abundances (filled circles) and standard deviations (open circles) and the corresponding model predictions (*y*-axis). The dashed red line represents $x = y$. (B) A comparison between OTU-OTU covariance $\langle \delta x_i \delta x_j \rangle$ as computed in the data and as predicted by the model. Absolute values are shown. (C) The inverse cumulative distribution representing the probability of observing an OTU abundance greater than a given abundance (*x*-axis) as computed from the data (black) and the corresponding model prediction (gray). The dashed lines show the best fit power law between relative abundances of $10^{-3}$ and $10^{-1}$. The slope of the power law is $-1.03$ in the data and $-1.12$ in the model. (D) The distribution of Shannon diversity of community compositions as observed in the data (gray) and computed from the model-generated communities (blue). (E) The distribution of Bray-Curtis dissimilarity between random pairs of community as well as the closest pairs of communities in the data (gray) and the model (blue). Distributions are smoothed using Gaussian kernel density estimation. (F) A scatter plot of Kullback-Leibler divergences between community composition in the data and the corresponding model fit (*x*-axis) and the log-likelihood of the embedded latent variables (*y*-axis). Data are shown for the training data from chicken cecum (black), testing data from chicken cecum (gray), testing data from chicken ileum (red), and testing data from chicken ileum embedded using a scrambled preference matrix (blue).

ileum of broiler chickens. The cecum and the ileum are distinct parts of the chicken's gastrointestinal track. Indeed, the abundances of microorganisms vary substantially between the two organs (42).

We split the cecum microbiome samples randomly into an 80% training and 20% testing split and trained the model only on the training data. As examples of dissimilar ecosystems, we used an additional testing set comprising ileal microbiomes of chickens. As seen in Fig. 3F, the species preferences toward effective resources are, indeed, universal. Unsurprisingly, the learnt species preferences embed testing data from the same organ site (cecum) with similar accuracy as the training data (black vs gray dots, *x*-axis). Surprisingly, the accuracy of embedding testing data from another organ using the same species preferences was quite similar to testing data from the same organ site (red vs black dots, *x*-axis). In contrast, the accuracy of embedding the testing data using a scrambled preference matrix was significantly lower (red vs blue dots, *x*-axis). Similarly, the inferred latent space locations of testing data microbiomes derived from ceca had a similar likelihood as the training data. In contrast, the latent space locations of ileal microbiomes had significantly lower likelihood (*y*-axis). This is consistent with our interpretation of the consumer resource framework. Specifically, while the OTUs have consistent embeddings between the ileum and the cecum, leading to reasonably accurate reconstruction of the ileal microbiomes, the abundances of effective resources and, therefore, the values of the fitted latent variables $z$s are likely to be markedly

different between the two organ sites. Consequently, the latent variables fitted for the ileums have significantly lower likelihood compared to the cecal latent variables. A similar result was observed when comparing embeddings of cattle from the Nordic Red breed using species preferences learnt from Holstein cows (Fig. S3).

Collectively, these results show that probabilistic modeling of the latent space defined by the C/R model generates microbial compositions that reproduce both universal statistics such as species abundance distributions and data set-specific statistics such as patterns of species covariances and α- and β-diversity metrics. Moreover, the learnt species-resource preferences are universal; the statistics of probability of observing unseen data and accuracy of embedding unseen data are similar to those of the training data. In other words, the C/R model-based generative model can sample compositions of feasible microbiome community structures, an essential feature of accurate generative models. We note that while we validated our model using three different host species, its application across diverse microbiomes and conditions, including environmental microbiomes (46) and plant microbiomes (47), will further test the robustness of our model beyond the gut of animal hosts.

## Generative modeling identifies host phenotypes that maximally constrain the microbiome

The coarse-grained latent space represents the history of effective resources. Therefore, it can be informative of host phenotypes that affect microbial growth. Indeed, recent work has suggested that many functions of microbial communities are coarse-grainable, that is, explained by a small number of variables (35). To test this, we investigated the quantitative relationship between phenotypes and associated microbiomes of the rumens of ~800 bovine hosts. Here, ~50 phenotypes were measured, including traits related to rumen chemistry (e.g., volatile fatty acids) and animal physiology (e.g., milk production, feed conversion efficiency) (Table S1).

Surprisingly, latent space embeddings inferred only using the microbiome accurately predicted most of the measured phenotypes using linear regression (Fig. S6). This suggests that the latent variables fitted to the C/R model encode information about the environmental variables experienced by the microbiome. Consequently, these effective resources may simultaneously capture microbial community structure and host phenotypes that influence microbial growth. To directly incorporate host phenotypes in our model, we employ a shared latent approach. Specifically, we require the latent variables to simultaneously describe the composition of microbiomes and host phenotypes. To that end, we introduce a new cost function:

$$C = \alpha \sum_{s,o} x_{so} \log q_{so} + (1-\alpha) \sum_{s,p} \left( m_{sp} - \sum_{k=1}^{K} z_{sk} C_{kp} \right)^2 \tag{5}$$

In equation 5, the first term captures the goodness of fit to the microbiome composition, and the second term is a model to fit the host phenotypes. Unlike the microbiome, the relationship between host phenotypes and latent variables cannot be mechanistically derived. Therefore, we take a data-driven approach. We approximate the host phenotypes using a linear low-rank model (see "Model description and fit to data"). We note that other functional forms, for example, feed forward neural networks can be employed as well. In equation 5, $1 \geq \alpha \geq 0$ decides the relative importance of the two terms in the minimization process. We fit equation 6 to the combined microbiome/phenotype data using $K = 24$ latent variables with $\alpha = 0.975$ (chosen to minimize the compromise in the accuracy of fitting the microbiome and phenotypes; Fig. S7). Next, as above, we trained a Gaussian mixture model on the shared latent space (see "Mixture Gaussian models"). In this case, sampling latent variables using the learnt distribution $p(z)$ generated combined microbiome/phenotype data that preserved covariances between the microbiome and the phenotypes (Fig. S8). Notably, the linear model employed in equation 5 may not be sufficient to model all host phenotypes from

the latent variables. However, at least in this case, nonlinear neural networks did not significantly improve the predictions (Fig. S6). Therefore, going forward, we employ linear models to connect host phenotypes to latent variables.

A central question in ecology is how strongly environmental parameters or ecosystem functions constrain the community composition. To answer this question in the context of host-associated microbiomes, we used generative modeling. Specifically, we sought to identify host phenotypes that, when set to specific values, constrain the number of possible microbial community structures consistent with said phenotypes. For example, host serum metabolomics may only weakly determine the microbiome composition, while host dietary habits and gut chemistry may have a more constraining impact. Indeed, our model finds that some phenotypes like rumen pH and propionate levels can significantly constrain the space of feasible community structures while others such as organic matter intake do not. Importantly, these dependencies are not always uniform across the range of host phenotypic states. To quantify these relationships, we generated a large number of *in silico* combinations of microbial communities and corresponding host phenotypes. For each phenotype, we evaluated the average Bray-Curtis dissimilarity among pairs of communities that corresponded to a narrow range of values of the phenotype. A low dissimilarity among communities corresponding to a particular range indicates that specifying the phenotype in that range narrows the space of feasible community compositions. We found that different phenotypes constrained community composition to a different degree. Figure 4A through C shows that specifying phenotypes such as levels of rumen short-chain fatty acids (propionate and isobutyrate) and rumen pH significantly constrained the microbiome composition in the rumen. In contrast, phenotypes such as organic matter intake and protein intake levels did not constrain the microbiome composition (Table S2). Notably, these analyses required *in silico* generation of thousands of feasible microbiome/host phenotype combinations and, therefore, cannot be performed using the data alone.

To verify the importance of specific phenotypes in narrowing plausible microbiome compositions, we investigated whether we could reconstruct the microbial composition in the rumens of hosts in the testing set using partial phenotypic information about the host. To do so, we deeply sampled from the mixture Gaussian model and found *in silico* samples that had phenotype values in close proximity of the data point of interest. We used the average microbiome composition of these *in silico* microbiomes as our microbiome prediction (see "Predicting microbiome composition from partial host phenotypic data"). We asked whether partial information about the phenotypes identified in Fig. 4A through C could identify the composition of the associated rumen microbiome. Importantly, the predictions made using partial phenotypic information were significantly more accurate compared to the dissimilarity between pairs of randomly picked samples in the testing data.

These results demonstrate that our generative model not only captures complex associations among microbial species (Fig. 3) but also the associations between the microbiome and corresponding host phenotypes (Fig. 4).

## Generative modeling quantifies context-specificity of host-microbiome interactions

Host-microbiome interactions are often context-specific, that is, the association between a taxon and a host phenotype may depend on the composition of the rest of the community. We exploit the generative capacity of our approach to identify subject-specific microbial correlates of methane production (normalized by dry matter intake). Methane production by bovines is a major contributor to greenhouse gasses and microbiome-based therapies to reduce methane production are a promising intervention avenue. To identify host-specific associations, we densely sampled the space of combined microbiome composition and host metadata using our generative model. Next, for each host, we identify *in silico* microbiomes that are in its proximity (evaluated using Bray-Curtis dissimilarity) and evaluate the correlation between microbial OTUs in

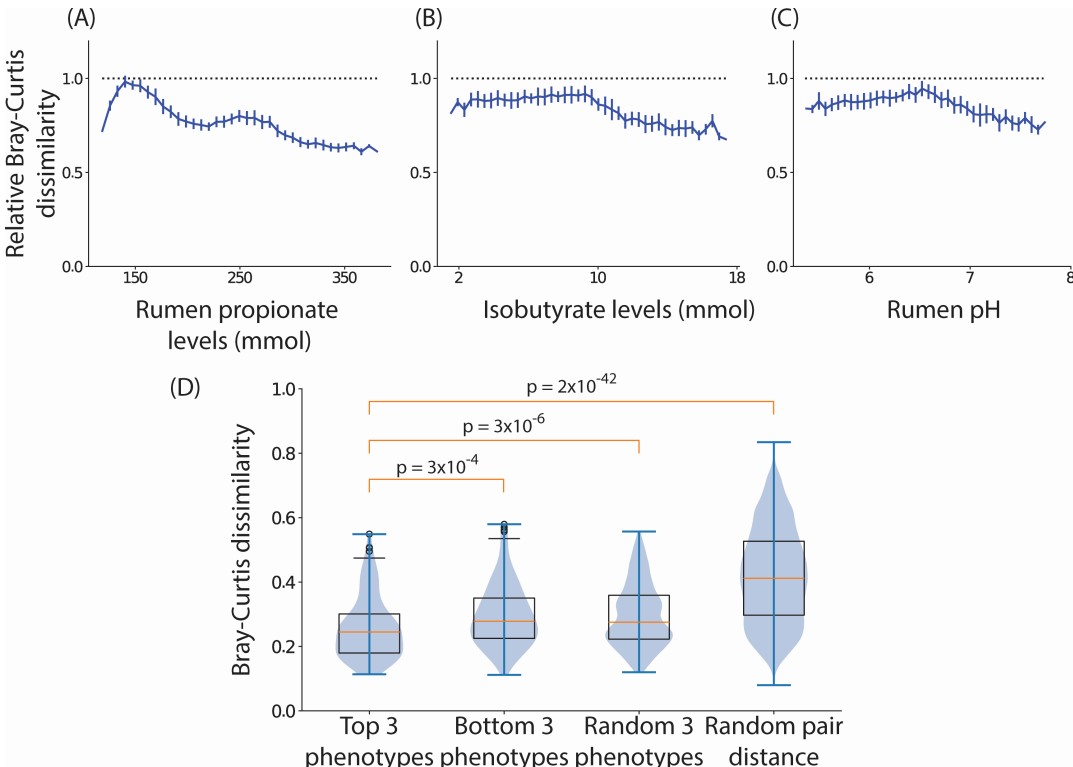

**FIG 4** Identifying host phenotypes that constrain the microbiome. (A, B, and C) The average Bray-Curtis dissimilarity (*y*-axis, normalized by the Bray-Curtis dissimilarity between random pairs of communities) among *in silico* bovine rumen microbial communities constrained to have a specified single phenotype (*x*-axis). The top three phenotypes that constrained community composition the most are shown. Error bars represent standard errors of the mean. (D) Box plot of Bray-Curtis dissimilarity between predicted microbiome composition and the measured composition using top three constraining phenotypes, bottom three least constraining phenotypes, three randomly chosen phenotypes, and random pairs of communities in the testing samples for bovine rumen communities.

this subset with methane production. As shown in Fig. 5, some OTUs associate with phenotypes in a universal manner, while others associate in a host-specific manner. To illustrate this more clearly, we look at two OTUs classified at the genus level as *Prevotella* and *Moraxellaceae ge* that have similar association strength with methane production when correlated across all samples. In contrast, our model predicts that the *Moraxellaceae ge* OTU correlates with methane across all hosts with similar magnitude, while the *Prevotella* OTU has a more heterogeneous association profile.

The phenotype/OTU associations calculated globally and on a per host basis (averaged over hosts) were correlated with each other. However, the average host-specific correlation was typically half as strong as the global correlation. Notably, some phenotypes were over-represented in host-specific local correlations. For example, of the top 100 host-specific correlated pairs, we found that certain phenotypes occurred more frequently than expected by chance. These include rumen propionate (22 out of 100, hypergeometric test $P = 4.08e{-}16$), rumen ammonia (11 out of 100, hypergeometric test $P = 1.80e{-}5$), fecal dry matter (11 out of 100, hypergeometric test $P = 2.18e{-}3$), and plasma creatinine (6 out of 100, hypergeometric test $P = 2.86e{-}2$).

## Designing microbiomes that achieve desired host phenotypes

The chief application of generative models is the design and identification of biological systems that possess user-desired properties, for example, proteins with specific enzymatic activity or structural folds (13). Here, we illustrate with two examples how our generative modeling can be used to identify host phenotypes and microbial ecosystem compositions that correspond to desired host states.

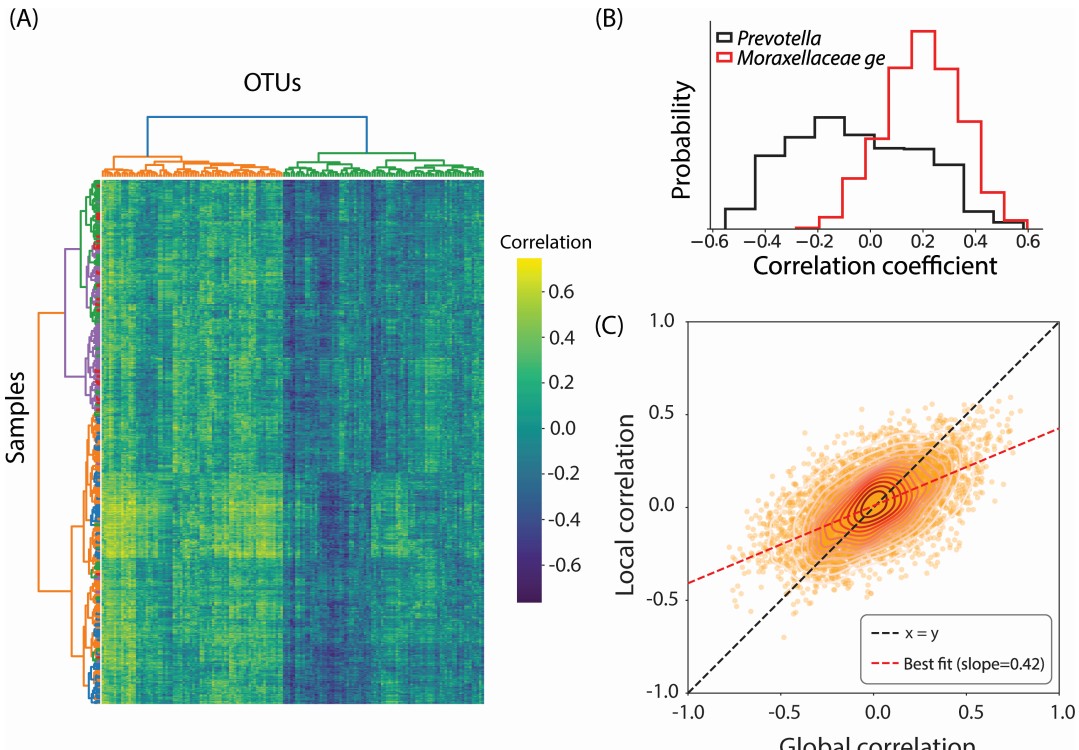

**FIG 5** Quantifying context-specificity of host-microbiome associations. (A) Pearson correlation coefficient between methane production (normalized by dry matter intake) and all OTUs evaluated in a sample-specific manner. Different colors in the sample-wise clustering tree represent different farms to which the cows belong. (B) Distribution of sample-specific Pearson correlation coefficients for two OTUs (*Prevotella* and *Moraxellaceae ge*). The global (across sample) Pearson correlation for abundance of either OTUs with methane production is ~0.15. (C) The mean host-specific (local) correlation of an OTU with a phenotype is plotted against the global correlation between that OTU and the phenotype, calculated *in silico*. The slope of the best fit line between local and global correlations is 0.42.

To validate that our generative model reproduces covariations observed in the data, we investigated the relationship between rumen propionate levels and rumen pH. Propionate production is associated with lower enteric methane emissions in ruminants due to its lower yield of molecular hydrogen (2). The pH of the rumen is an emergent property arising from the complex acid/base balance of multiple chemicals. Notably, rumen pH negatively correlates with propionate levels (Spearman $r = -0.40$, $P = 1.5 \times 10^{-31}$), which was not true of all short-chain fatty acids (e.g., Spearman $r = 0.42$, $P = 3.2 \times 10^{-34}$ for butyrate and Spearman $r = 0.10$, $P = 4.5 \times 10^{-3}$ for acetate). Therefore, we sought to characterize host phenotypic states that corresponded with simultaneous high rumen pH and high propionate levels. To achieve this, we performed a biased sampling of the latent space using a Markov chain Monte Carlo approach (see "Biased sampling of the latent space to identify microbiome compositions corresponding to desired host states"; red contours in Fig. 5A). Notably, these hosts were not observed in the training population (black contours in Fig. 5A). Next, we investigated how this shift in propionate and pH changed the levels of other short-chain fatty acids. Surprisingly, levels of both acetate and butyrate, which positively correlated with pH, dropped in the biased samples. While not apparent from the data, these results make physiological sense; increasing propionate concentration may be compensated by a decrease in acetate and butyrate to maintain mass balance and rumen pH.

Our model can also identify the relationship between host phenotypes and the corresponding composition of the microbiome. Cows fed high amounts of starch often have acidic rumens due to its rapid fermentation compared to fiber-rich diets. This may lead to sub-acute ruminal acidosis (SARA), an undesirable condition that leads to

lowered economic productivity (48). Indeed, in the data set analyzed, starch intake was negatively correlated with pH (Spearman $r = -0.53$, $P = 1.6 \times 10^{-58}$). Notably, there were no hosts in the original data set who could maintain a high pH on a starch-rich diet (black contours in Fig. 5C). Given that the pH directly affects microbial growth and can also be controlled by it, we investigated microbial ecosystem compositions that would be present at a high rumen pH in hosts that have a high starch intake using biased sampling of the latent space (see "Biased sampling of the latent space to identify microbiome compositions corresponding to desired host states"; red contours in Fig. 6C). The rest of the host phenotypes (and the composition of the microbiome) were not specified but automatically sampled according to the biased distribution. The microbiome compositions obtained were significantly different for these desired host phenotypes (Fig. 5D). Yet, these *in silico* microbiomes have statistical properties (e.g., α- and β-diversity) that are quite similar to real microbiomes, suggesting that these microbiomes may exist in nature (Fig. S9). Two OTUs were consistently present in higher abundance in the *in silico* biased samples with high starch intake and a high pH (Table S3). These include an OTU from the genus *Ruminobacter* (1 out of 2, hypergeometric test $P = 0.025$) and an OTU classified as *Succinivibrionaceae_UCG-002*. Both OTUs belonged to the family *Succinivibrionaceae* (2 out of 12, hypergeometric test $P = 0.005$). Notably, *Ruminobacter* bacteria are known to degrade starch (47). However, their ability to do so while maintaining a high rumen pH is not fully explored.

These studies show that our model can sample and thereby design host-microbiome metacommunities with user-specified host phenotypes.

## DISCUSSION

Rationalizing the observed variation in complex and high-dimensional biological systems using mechanistic modeling is seldom possible. At the same time, we can now collect large amounts of data. This has allowed building data-driven generative models that describe possible variations in biological systems. Unfortunately, except for a few examples, studies on host-associated microbiomes usually operate with low sample sizes. Moreover, the context of the microbiome sample (abiotic environment or the phenotypic states of the host organism) is of paramount importance in deciding ecosystem composition. This context dependence likely contributes to a low reproducibility of microbiome studies (49).

In this work, we presented a generative model based on the consumer/resource framework that can model the simultaneous variation in host-associated microbiomes and host's phenotypes. Notably, our model can assign probabilities to specific communities and phenotypes. This allows us to identify realistic communities that are hypothesized to correspond to desired host states.

Importantly, although our model allows us to sample realistic microbial communities and associated host phenotypes, it does not provide a causal connection between the two. We have shown that certain host traits such as organ pH, nutrient intake, or the concentration of specific metabolites are associated with more specific microbial communities within certain trait value ranges. This does not necessarily mean that those parameters can be used to drive the microbiome toward a particular state although these are useful hypotheses. Instead, the ability to generate an arbitrary number of realistic *in silico* samples facilitates the study of complex communities by enabling data stratification beyond what is experimentally possible, even to cases that are seldom observed in experimental data sets. As a design tool, the communities obtained by sampling from the model represent target communities for microbiome manipulation strategies including pre- and probiotic treatments, antibiotics, microbiome transplants, phage therapies, etc. Knowing what these targets look like should provide clues as to how to achieve them. Notably, however, the model can easily be made causal by incorporating a temporal direction, for example, by training the model on pre- and post-intervention (drugs, probiotics, etc.) microbiomes.

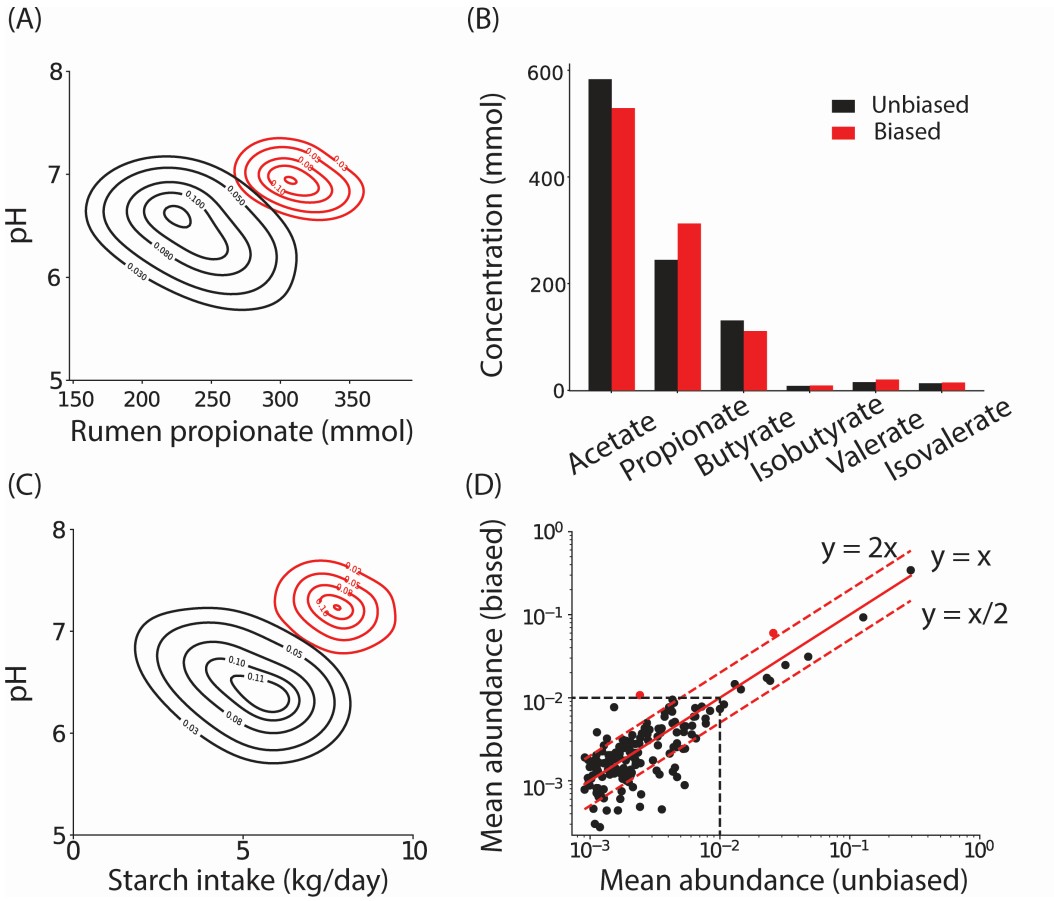

**FIG 6** Designing microbiomes for desired host phenotypic states. (A) Contour plots of the two-dimensional histograms of rumen pH and rumen propionate levels. Black contours represent *in silico* communities sampled from the inferred distribution over latent variables. Red contours represent the biased sampling of latent variables which encourages high propionate levels and high rumen pH. (B) Concentrations of volatile fatty acids in unbiased (black) and biased (red) samples. (C) Contour plots of the two-dimensional histograms of rumen pH and daily starch intake levels. Black contours represent *in silico* communities sampled from the inferred distribution over latent variables. Red contours represent the biased sampling of latent variables which encourages high levels of rumen pH while maintaining a high starch intake. (D) Mean relative abundances of OTUs in the unbiased samples (*x*-axis) and the biased samples (*y*-axis). The OTUs with high abundance in the biased samples that also show a significant increase from unbiased samples are colored in red.

We demonstrated that species preferences learned by our model can be used to model microbiome compositions across different communities comprised of the same species (Fig. 3; Fig. S3). While this indicates that the latent space explains microbial abundances across biological contexts, a limitation of our approach is that the latent space may not always be interpretable as specific resources or environmental parameters. First, by coarse-graining the environment using the latent space to reproduce microbiome compositions, the latent space represents emergent effective resource combinations and not individual resources. Second, by defining latent variables as time-integrals of effective resources, their values need not necessarily match measured resource abundances at a given moment in time. To increase the interpretability of the latent variables, known species preferences toward specific resources could be hard coded into the model, with the corresponding latent variables indicating the variation of said resources across samples. The latent variables may also be interpreted by studying their association with measured host metadata, for example, by studying the coefficients of the matrix $C$ in equation 5. Nevertheless, this approach needs to be validated by demonstrating the similarity of mappings between latent variables and phenotypes across independent and diverse data sets. Our analyses show that even in the absence of a one-to-one correspondence between latent variables and measurable variables, the

generative ability of our model enables studying the relationship between microbiomes and host phenotypes.

There are several ways to generalize the current framework. First, we learnt the latent space embeddings $z$s of individual hosts first and then built a probabilistic model $p(z)$ that captured the variation in the latent variables. This two-step process can be combined using Expectation-Maximization like algorithms that simultaneously learn embedding for data and model the corresponding mixture Gaussian distribution. Second, while the dependence of microbial abundances on latent variables was mechanistically derived, the linear relationship between host phenotypes and latent variables was *ad hoc*. This linear model can in principle be improved by incorporating more complex relationships between the phenotypes and the latent variables, for example, those modeled using artificial neural networks. Third, in the current approach, we built a combined latent space model for host phenotypes and the associated microbiomes. Similarly, phenotypic information about microorganisms, for example, consumer/resource preferences inferred using genome-scale metabolic models (50) or species genomic content, can also be incorporated in our framework by requiring the species-specific preferences $\theta_{ko}$ to simultaneously model species abundances in microbiomes and species phenotypes using previously developed latent variable models (17).

While this model was developed for host-associated microbiomes, it can be used to integrate different types of sequencing and phenotypic information not only on animal hosts and corresponding microbial ecosystems but also on cells and their phenotypic properties and potentially in a more general setting when both sequencing and phenotypic information is available for the same entities.

We believe that the framework laid out in the manuscript will be of broad use to microbiome engineers and scientists alike.

## MATERIALS AND METHODS

### Model description and fit to data

#### *Details of the latent variable model and performance*

The consumer/resource (C/R) model (28) has been a popular model in understanding species dynamics in natural and artificial ecosystems. The C/R model imagines that species abundances fluctuate according to their dependence on resources. We have

$$\frac{1}{n_o}\frac{dn_o}{dt} = \mu_o(t) = \sum_{k=1}^{K} r_k(t)\theta_{ko} - \delta_o \qquad (6)$$

and

$$\frac{dr_k}{dt} = J_k(t) - r_k(t)\sum_{o=1}^{0} n_o(t)\theta_{ko}\gamma_{ko}. \qquad (7)$$

In equations 6 and 7, $n_o(t) \geq 0$ are abundances of consumers (microbial species) $o$, $\mu_o(t)$ is the time-dependent growth rate, $r_k(t) \geq 0$ is the abundance of resource $k$, $\theta_{ko} \geq 0$ are species-preferences for resources, $\gamma_{ko} \geq 0$ are resource consumption efficiencies, and $J_k(t)$ are net inflow rates of resources.

With these assumptions, we can formally integrate equation 1 to obtain

$$n_o(t) \approx \exp\left(\int_0^t \mu_o(\tau)d\tau\right) = \exp\left(-\sum_{k=1}^{K} z_k\theta_{ko} - \delta_o t\right) \qquad (8)$$

where we recognize

$$z_k = -\int_0^t r_k(\tau)d\tau \tag{9}$$

as latent variables that represent the complex temporal dynamics of resources. We note that the death/dilution term can be incorporated into the latent variable representation by identifying $z_{K+1} = t$ (the relevant time scale for the establishment of the current community composition) and $\theta_{K+1,o} = \delta_o$. Shifting the indices from $K+1$ to $K$ for simplicity, the relative abundances of organisms $q_o$ are simple functions of the latent variables:

$$q_o = \frac{1}{\Omega}\exp\left(-\sum_{k=1}^{K} z_k\theta_{ko}\right). \tag{10}$$

Equation 10 can be used either in a longitudinal context where latent variables $z_k$ evolve over time inside a single host or in a cross-sectional context where we imagine that microbiomes associated with similar hosts (e.g., same host organism) couple to the latent variables in the same manner and that latent variables represent snapshots of individual hosts.

Here, we describe in detail the most general formulation of our approach; to model a collection of host-associated ecosystems along with the phenotypes of the host. We start with the measurement of relative abundances of species (or operational taxonomic units, OTUs) in microbial ecosystems associated with hosts. We denote these abundances as $x_{so}$, where $s \in [1, S]$ is the index of the subject and $o$ is an organism. Additionally, we assume that measurements of $P$ phenotypic metadata $m_{sp}(p \in [1, P])$ that can potentially affect the microbiome are also measured.

### Fitting the model to microbiome data

We write the model-predicted composition of the ecosystems as

$$q_{so} = \frac{1}{\Omega_s}\exp\left(-\sum_{k=1}^{K} z_{sk}\theta_{ko}\right). \tag{11}$$

Since the latent variables are related to the properties of the ecosystem and the host, we further hypothesize that the host's phenotypic metadata explain the latent variables. Unfortunately, however, the relationship between latent variables and host phenotypes cannot be derived using mechanistic models. To that end, we use data-driven methods to capture this dependence. Specifically, we use a generalized low rank linear model:

$$m_{sp} \approx \sum_{k=1}^{K} z_{sk}L_{kp}. \tag{12}$$

where $g(\cdot)$ is a user-specified function. We use $g(\cdot) = I(\cdot)$ (identity function). Importantly, we require that the same host-specific latent variables explain the composition of all ecosystems as well as the host's phenotypic information. To that end, we write a combined cost function:

$$C = (1-\alpha)C_m + \alpha C_e \tag{13}$$

where for the bovine rumen data, $C_m$ is the squared $L_2$ loss between the host metadata and the corresponding model prediction. $C_e$ is the Kullback-Leibler divergence between the measured microbial composition in the ecosystem and the corresponding model prediction. $0 \leq \alpha \leq 1$ is a scaling factor that determines the relative importance of the two terms. When fitting only the microbiome data, we set $\alpha = 0$. We have the derivatives:

10.1128/msystems.01068-24 15

$$\frac{\partial C_m}{\partial z_{sk}} = -2\sum_p \left( m_{sp} - \left( \sum_{k=1}^K z_{sk} L_{kp} \right) \right) L_{kp} \tag{14}$$

and

$$\frac{\partial C_m}{\partial L_{kp}} = -2\sum_s \left( m_{sp} - \left( \sum_{k=1}^K z_{sk} L_{kp} \right) \right) z_{sk}. \tag{15}$$

The derivatives for the cost $C_e$ are

$$\frac{\partial C_e}{\partial z_{sk}} = \sum_0 (x_{so} - q_{so}) \theta_{ko} \tag{16}$$

and

$$\frac{\partial C_e}{\partial \theta_{ko}} = \sum_s (x_{so} - q_{so}) z_{sk} \tag{17}$$

We infer the model parameters using gradients in equations 14–17 and constant step-size gradient descent with learning rate $\eta$ ranging between $\eta = 10^{-3} - 10^{-5}$ depending on the data set. The learning rate was set to ensure that the loss decreased smoothly with the number of iterations. We stopped the learning process when the sum of $L_2$ norms of the gradients divided by $L_2$ norms of the corresponding matrices was less than $10^{-2}$. Notably, the inference problem is convex in $z$s, $\theta$s, and $L$ when the other parameters are kept constant (51). Therefore, the inferred matrices were robust (up to a rotation) to hyperparameters related to learning.

## Details of the data sets

Three data sets were used in our study that spanned multiple animal hosts. We used data collected on (i) bacterial microbiome compositions in the bovine rumen as well as physiological information about the bovine host, (ii) bacterial ecosystems in chicken ceca, (iii) fecal bacterial microbiomes in humans. Below, we provide details of individual data sets.

### Bovine rumen microbiome and host physiology

The data on rumen microbiomes and host physiology of Holstein cows were downloaded from the paper by Wallace et al. (37). For microbiome profiling, raw sequencing reads targeting the bacterial 16S rRNA gene were downloaded from SRA and processed with DADA2 (v.1.12.1) (52) to generate a matrix of read counts per sample at the level of amplicon sequence variants (ASVs). ASVs were clustered into 97% identity OTUs using CD-HIT (53), and OTU-level relative abundances were calculated from the ASV counts using the corresponding clusters. The assignTaxonomy method of DADA2 was used to assign genus-level taxonomic labels to the OTUs based on the Silva v. 138 database (54). We only included bacterial OTUs whose average relative abundance was larger than 0.1%. This cutoff was based on our previous work where we had shown that OTU abundance variation below 0.1% is likely to be dominated by technical noise. In the end, we had data on relative abundances of 156 bacterial OTUs in 790 cows. An additional OTU represented the combined abundance of all OTUs whose average abundance was below 0.1%. For training/testing purposes, the data were randomly split into a 80% training set and a 20% testing set. The OTUs identified in Holstein cows were also used to define the microbiomes of Nordic red cows. There were 156 bovine hosts belonging to the Nordic red species.

### Chicken microbiomes

The data on chicken cecal microbiomes were downloaded from the paper by Johnson et al. (38). Raw 16S rRNA amplicon sequences were processed as described above. As mentioned above, we only included OTUs whose average relative abundance was larger than 0.1%. In the end, we had a total of 425 host samples and 147 OTUs. As mentioned above, one OTU represented the combined abundance of all OTUs whose average abundance was below 0.1%.

### Human fecal microbiome

Metagenomics-derived genus-level microbial abundances were obtained from the Inflammatory Bowel Disease database (39). We considered 127 samples from the healthy controls. As above, genera with average relative abundances across samples lower than 0.1% were aggregated into a single taxon, yielding a total of 95 bacterial genera analyzed.

### Mixture Gaussian models

In our model, the compositions of the microbiomes and host metadata are direct functions of the latent space variables. We make our model generative by approximating a distribution over the latent space. To that end, we fit a mixture multivariate Gaussian model using Python's sklearn.mixture.GaussianMixture to the inferred latent space descriptors. The number of Gaussians in the mixture model is varied between 1 and 5 and the mixture with the lowest Bayesian information criterion (BIC). Given that fitting a mixture Gaussian model typically leads to a locally optimum solution, the number of Gaussians in the mixture and the mean values and covariances matrices of the mixtures may change from fit to fit. Therefore, we ran the procedure 100 times and chose the Gaussian mixture model with the lowest BIC.

### Predicting microbiome composition from partial host phenotypic data

To identify microbiome compositions corresponding to a specified partial list of host phenotypes, we deeply sampled the inferred mixture Gaussian distribution and retained the samples with phenotypes of interest with values in proximity to the host values (within two-tenths of the standard deviations of the measured phenotypes). We use the mean composition of the communities identified to have the required phenotype values as our predicted microbiome composition. The Bray-Curtis dissimilarity between this predicted microbiome composition and the host microbiome composition was used as the error/distance measure.

### Biased sampling of the latent space to identify microbiome compositions corresponding to desired host states

To identify microbiome compositions corresponding to desired host phenotypes, we sampled latent variables using the inferred mixture Gaussian distribution amended with a biasing term. Specifically, we modified the log probability of latent variables as

$$\log p(z) = \log p_0(z) - \gamma(\mathrm{st} - 3)^2 - \gamma(\mathrm{pH} - 3)^2 + \mathrm{const} \qquad (18)$$

In equation 18, $p_0(z)$ is the inferred mixture Gaussian model, st denotes $z$-scored starch intake, and pH denotes $z$-scored rumen pH. The starch intake and the pH were predicted from the latent variables using the inferred linear model (equation 12). $\gamma = 0$ corresponded to the unbiased simulation and biasing was achieved by setting $\gamma = 2$. To sample the latent space, we randomly chose 100 starting points from $p_0(z)$ and performed a Markov chain Monte Carlo simulation with the modified probability. Once the latent variables are sampled, microbiome compositions are predicted using equation 11.

## ACKNOWLEDGMENTS

P.D. is supported by NIGMS grant R35GM142547. P.D. and K.S. are supported by the Bill and Melinda Gates Foundation grant INV-062437 to BiomEdit.

## AUTHOR AFFILIATIONS

[1]Computational Sciences, BiomEdit, LLC., Fishers, Indiana, USA
[2]Department of Biomedical Engineering, Yale University, New Haven, Connecticut, USA
[3]Discovery Research, Elanco Animal Health, Greenfield, Indiana, USA
[4]Department of Physics, University of Florida, Gainesville, Florida, USA
[5]Systems Biology Institute, Yale University, West Haven, Connecticut, USA

## PRESENT ADDRESS

Madan Krishnamurthy, Department of Biomedical informatics, University of Colorado, Aurora, Colorado, USA

Lukas Herron, Institute of Physical Sciences and Technology, University of Maryland, College Park, Maryland, USA

## AUTHOR ORCIDs

Germán Plata http://orcid.org/0000-0002-6470-7748
Purushottam Dixit http://orcid.org/0000-0003-3282-0866

## FUNDING

| Funder | Grant(s) | Author(s) |
| --- | --- | --- |
| HHS | NIH | National Institute of General Medical Sciences (NIGMS) | R35GM142547 | Karthik Srinivasan |
| | | Purushottam Dixit |
| Bill and Melinda Gates Foundation (GF) | INV-062437 | Germán Plata |
| | | Karthik Srinivasan |
| | | Purushottam Dixit |

## AUTHOR CONTRIBUTIONS

Germán Plata, Conceptualization, Data curation, Formal analysis, Funding acquisition, Investigation, Methodology, Project administration, Resources, Software, Supervision, Validation, Visualization, Writing – original draft, Writing – review and editing | Karthik Srinivasan, Conceptualization, Data curation, Formal analysis, Investigation, Methodology, Resources, Software, Validation, Visualization, Writing – review and editing | Madan Krishnamurthy, Data curation, Formal analysis, Investigation | Lukas Herron, Data curation, Formal analysis, Investigation | Purushottam Dixit, Conceptualization, Data curation, Formal analysis, Funding acquisition, Investigation, Methodology, Project administration, Resources, Software, Supervision, Validation, Visualization, Writing – original draft, Writing – review and editing

## DATA AVAILABILITY

No new experimental data were generated for this manuscript. All scripts and simulation data are available on github (https://github.com/karthik-yale/host_microbe) and Zenodo (https://zenodo.org/records/13998700).

## ADDITIONAL FILES

The following material is available online.

### Supplemental Material

**Supplemental Figures (mSystems01068-24-s0001.pdf).** Figures S1 to S9.

**Legends (mSystems01068-24-s0002.docx).** Legends for supplemental material.
**Supplemental Tables (mSystems01068-24-s0003.xlsx).** Tables S1 and S2.

## Open Peer Review

**PEER REVIEW HISTORY (review-history.pdf).** An accounting of the reviewer comments and feedback.

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
