## [Reviewer comments · mSystems]

Designing host-associated microbiomes using the consumer/resource model

German Plata, Karthik Srinivasan, Lukas Herron, Madan Krishnamurthy, and Purushottam Dixit

Corresponding Author(s): Purushottam Dixit, Yale University

Review Timeline:

Submission Date:	August 22, 2024
Editorial Decision:	October 15, 2024
Revision Received:	November 4, 2024
Accepted:	November 6, 2024

Editor: John Gibbons

Reviewer(s): Disclosure of reviewer identity is with reference to reviewer comments included in decision letter(s). The following individuals involved in review of your submission have agreed to reveal their identity: Leo Lahti (Reviewer #3)

Transaction Report:

DOI: <https://doi.org/10.1128/msystems.01068-24>

Re: mSystems01068-24 (Designing host-associated microbiomes using the consumer/resource model)

Dear Prof. Purushottam Dixit:

Thanks for your thoughtful revision. The major barrier to accepting this manuscript as it currently stands is due to reproducibility. As one reviewer pointed out, "The Jupyter notebooks on the GitHub repository rely on hardcoded file paths that point either to missing data files or to inaccessible locations on one of the authors' personal file system. This currently prevents independent execution and verification, which is essential for publication." Another review recommends establishing a permanent DOI for your code and dataset. I also agree with both points as reproducibility is critical. Reviewer 1 also brought up an important concern about the interpretability of latent variables that should be directly addressed in your revision.

Revision Guidelines

Sincerely,
John Gibbons
Editor
mSystems

Reviewer #2 (Comments for the Author):

This revised manuscript addresses the previous concerns effectively and represents a valuable contribution to the field of microbiome research. The authors have significantly improved the clarity, reproducibility, and scope of their work. The addition of the "dissimilar microbial community", discussion on causality, and exploration of nonlinear models significantly strengthens the paper. Unfortunately, despite the improvements in the codebase, a critical reproducibility issue remains. The Jupyter notebooks on the GitHub repository rely on hardcoded file paths that point either to missing data files or to inaccessible locations on one of the authors' personal file system. This currently prevents independent execution and verification, which is essential for publication.

Regarding the major points that were raised:

1. Data and code: The inclusion of appropriate documentation and an open-source license in the GitHub repository is commendable. However, the critical reproducibility issue identified previously remains a significant barrier to publication. Please add all the required data files and please proceed to test the updated codebase in a fresh environment to verify that you are not missing any piece. Some examples of needed corrections or missing data are (not a comprehensive list):

___ A) In the GitHub repository documentation/README:

___ * Please correct the title of the README ("micorbiomes" to microbiomes).

___ * Add the right cloning instruction for cloning the repository; please add real data instead of the placeholders, e.g.,
`host_microbe`
` instead of `your_repository_name`.

___ * Despite the conda installation and before being able to launch Jupyter Notebook (error `Jupyter command `jupyter-notebook` not found`) it may be necessary to manually install further dependencies by using `pip install jupyter notebook`

___ B) Notebook for Fig 1: Please correct over-indenting in the first indented line of the class fig_1 as it generates a fatal error.

___ C) Notebook for Fig 2: Missing path and files `fig_2_files/cow_z.csv` and `fig_2_files/cow_theta.csv`.

___ D) Notebook for Fig 3:

___ * Please correct inaccessible locations out of the repository: `/home/ks2823/palmer_scratch/my_object.pkl`.

___ * Missing path and files: `fig_3_files/combined_latents.csv`, `fig_3_files/theta.csv`, `fig_3_files/metadata_theta.csv`.

___ E) Notebook for Fig 4:

___ * Missing file `data_files/all_phens_covs.json`

___ * Please correct inaccessible locations out of the repository:

`/home/ks2823/my_CRM/Data/three_kingdoms_python_friendly.pkl` and `/home/ks2823/palmer_scratch/my_object.pkl`.

___ F) Notebook for Fig 5:

___ * The directory `fig_3_files` is missing (see D above) and it's needed for this notebook too.

___ * Missing file `data_files/cow_survived_indices.csv`.

2. Interpretability of Latent Variables: While the authors have attempted to address the concerns regarding interpretability by linking the latent space to phenotypic metadata, their argument is not fully convincing. Low dimensionality does not intrinsically ensure interpretability, particularly when the mapping between latent variables and original microbiome features remains unclear and there is a lack of demonstration of generalizability to new data. Linking the latent space to phenotypes is essential, but demonstrating that the learned latent space and phenotype mappings generalize to independent datasets is crucial for supporting their claim of interpretability. Exploring and visualizing generalizable relationships between the latent variables and original microbiome features (e.g., through feature importance or contribution analysis) would clarify what the latent variables represent biologically. Comparing the interpretability of the proposed approach to dimensionality reduction techniques in the context of host phenotypes may provide valuable context. Alternatively, a more explicit acknowledgment of the limitations of interpretability and a discussion of future research directions to address them would strengthen the paper.

3. Validation on diverse microbiomes: The cross-organ validation (chicken ceca to chicken ileum) and the inclusion of three different host species are valuable additions, demonstrating the model's potential for broader applicability. Future testing on more diverse datasets and conditions is warranted to confirm robustness.

4. Exploring causal relationships: The authors' acknowledgment of the model's correlational nature and their proposed strategies for incorporating temporal data to explore causality are important additions that improve the paper.

5. Incorporating nonlinear relationships: The exploration of nonlinear models and the justification for using a linear model in this specific context are satisfactory. The authors' clear explanation for the lack of comparable models for simultaneous microbiome and phenotype generation is also appreciated.

Regarding the minor points that were raised:

* Taxa Exclusion: This point was adequately addressed and the authors additional analysis show that the model is relatively insensitive to specific abundance cutoffs.

* Model Fitting Details: Regarding authors' reply, please see the comments above about the Jupyter notebooks, which have several problems. The provided details about hyperparameters and convergence criteria are appreciated.

* There is an additional minor point after the revision: The question "Specifically, we sought to identify which host phenotypes constrain the space of possible microbial community structures?" could be rephrased for greater clarity.

Reviewer #3 (Comments for the Author):

I would like to thank for addressing the previous review suggestions.

I just have two minor comments remaining:

1) It would be advisable to associate permanent DOI with the code and data repository (e.g. via Zenodo), and/or add this as a supplementary. There is no guarantee that the proprietary Github repository will remain available in the long run. This would guarantee that the exact code version used in this manuscript will be preserved permanently.

<https://docs.github.com/en/repositories/archiving-a-github-repository/referencing-and-citing-content#issuing-a-persistent-identifier-for-your-repository-with-zenodo>

2) Spell-check for the manuscript and code repository (e.g. micorbiome -> microbiome etc.)

1 Dear Editors,

2

3 We thank the two reviewers for their comments. We apologize for the gaps in our code that made it
4 hard to re-create our analysis. We have now reworked our GitHub repository and confirmed its
5 reproducibility via independent testing. A DOI for a copy of the code in Zenodo has also been included.
6 We have also addressed issues raised by reviewer #2 regarding interpretability of the latent variables.
7 Finally, we have addressed the minor issues (typos etc.).

8

9 Thank you,

10 Purushottam Dixit and Germán Plata

11

12 Reviewer #2 (Comments for the Author):

13

14 This revised manuscript addresses the previous concerns effectively and represents a valuable
15 contribution to the field of microbiome research. The authors have significantly improved the clarity,
16 reproducibility, and scope of their work. The addition of the "dissimilar microbial community",
17 discussion on causality, and exploration of nonlinear models significantly strengthens the paper.
18 Unfortunately, despite the improvements in the codebase, a critical reproducibility issue remains. The
19 Jupyter notebooks on the GitHub repository rely on hardcoded file paths that point either to missing
20 data files or to inaccessible locations on one of the authors' personal file system. This currently prevents
21 independent execution and verification, which is essential for publication.

22

23 We apologize for the lack of transparency in our GitHub code. We have now modified it to make it
24 reproducible. The two first authors have tested it independently to ensure reproducibility. We hope that
25 these changes satisfy the reviewer's concerns.

26

27 Regarding the major points that were raised:

28

29 1. Data and code: The inclusion of appropriate documentation and an open-source license in the GitHub
30 repository is commendable. However, the critical reproducibility issue identified previously remains a
31 significant barrier to publication. Please add all the required data files and please proceed to test the
32 updated codebase in a fresh environment to verify that you are not missing any piece. Some examples
33 of needed corrections or missing data are (not a comprehensive list):

34

35 __ A) In the GitHub repository documentation/README:

36 ____ * Please correct the title of the README ("micorbiomes" to microbiomes).

37

38 We have fixed this typo.

39

40 ____ * Add the right cloning instruction for cloning the repository; please add real data instead of the
41 placeholders, e.g., `host_microbe` instead of `your_repository_name`.

42

43 We have added cloning instructions and have added paths to real data.

44

45 ____ * Despite the conda installation and before being able to launch Jupyter Notebook (error `Jupyter
46 command `jupyter-notebook` not found`) it may be necessary to manually install further dependencies
47 by using `pip install jupyter notebook`

48

49 We have fixed this.

50

51 __ B) Notebook for Fig 1: Please correct over-indenting in the first indented line of the class `fig_1` as it
52 generates a fatal error.

53

54 We have fixed this.

55

56 __ C) Notebook for Fig 2: Missing path and files ``fig_2_files/cow_z.csv`` and ``fig_2_files/cow_theta.csv``.

57

58 We have fixed this.

59

60 __ D) Notebook for Fig 3:

61 ____ * Please correct inaccessible locations out of the repository:
62 ``/home/ks2823/palmer_scratch/my_object.pkl``.

63

64 We have fixed this.

65

66 ____ * Missing path and files: ``fig_3_files/combined_latents.csv``, ``fig_3_files/theta.csv``,
67 ``fig_3_files/metadata_theta.csv``.

68

69 We have fixed this.

70

71 __ E) Notebook for Fig 4:

72 ____ * Missing file ``data_files/all_phens_covs.json``

73

74 We have fixed this.

75

76 ____ * Please correct inaccessible locations out of the repository:
77 ``/home/ks2823/my_CRM/Data/three_kingdoms_python_friendly.pkl`` and
78 ``/home/ks2823/palmer_scratch/my_object.pkl``.

79

80 We have fixed this.

81

82 __ F) Notebook for Fig 5:

83 ____ * The directory ``fig_3_files`` is missing (see D above) and it's needed for this notebook too.

84

85 We have fixed this.

86

87 ____ * Missing file ``data_files/cow_survived_indices.csv``.

88 We have fixed this.

89

90

91 2. Interpretability of Latent Variables: While the authors have attempted to address the concerns
92 regarding interpretability by linking the latent space to phenotypic metadata, their argument is not fully
93 convincing. Low dimensionality does not intrinsically ensure interpretability, particularly when the
94 mapping between latent variables and original microbiome features remains unclear and there is a lack
95 of demonstration of generalizability to new data. Linking the latent space to phenotypes is essential, but
96 demonstrating that the learned latent space and phenotype mappings generalize to independent

97 datasets is crucial for supporting their claim of interpretability. Exploring and visualizing generalizable
98 relationships between the latent variables and original microbiome features (e.g., through feature
99 importance or contribution analysis) would clarify what the latent variables represent biologically.
100 Comparing the interpretability of the proposed approach to dimensionality reduction techniques in the
101 context of host phenotypes may provide valuable context. Alternatively, a more explicit
102 acknowledgment of the limitations of interpretability and a discussion of future research directions to
103 address them would strengthen the paper.

104

105 We thank the reviewer for raising this very important concern. We now acknowledge this limitation in
106 an extended discussion on the latent variable interpretability. We agree with the reviewer's suggestion
107 that interpretability of the latent space must be demonstrated using independent datasets, we have
108 included this along with other suggestions to help with interpretability. (lines 498-514).

109

110 3. Validation on diverse microbiomes: The cross-organ validation (chicken ceca to chicken ileum) and the
111 inclusion of three different host species are valuable additions, demonstrating the model's potential for
112 broader applicability. Future testing on more diverse datasets and conditions is warranted to confirm
113 robustness.

114

115 We thank the reviewer for the comment. We have now added some text to stress that further testing on
116 diverse datasets is needed (lines 320-323).

117

118 4. Exploring causal relationships: The authors' acknowledgment of the model's correlational nature and
119 their proposed strategies for incorporating temporal data to explore causality are important additions
120 that improve the paper.

121

122 We thank the reviewer for the comment.

123

124 5. Incorporating nonlinear relationships: The exploration of nonlinear models and the justification for
125 using a linear model in this specific context are satisfactory. The authors' clear explanation for the lack
126 of comparable models for simultaneous microbiome and phenotype generation is also appreciated.

127

128 We thank the reviewer for the comment.

129

130 Regarding the minor points that were raised:

131

132 * Taxa Exclusion: This point was adequately addressed and the authors additional analysis show that the
133 model is relatively insensitive to specific abundance cutoffs.

134

135 We thank the reviewer for the comment.

136

137 * Model Fitting Details: Regarding authors' reply, please see the comments above about the Jupyter
138 notebooks, which have several problems. The provided details about hyperparameters and convergence
139 criteria are appreciated.

140

141 We thank the reviewer for the comment.

142

143 * There is an additional minor point after the revision: The question "Specifically, we sought to identify
144 which host phenotypes constrain the space of possible microbial community structures?" could be
145 rephrased for greater clarity.

146
147 We have now fixed this text (lines 363-365).

148
149 Reviewer #3 (Comments for the Author):

150
151 I would like to thank for addressing the previous review suggestions.

152
153 I just have two minor comments remaining:

154
155 1) It would be advisable to associate permanent DOI with the code and data repository (e.g. via Zenodo),
156 and/or add this as a supplementary. There is no guarantee that the proprietary Github repository will
157 remain available in the long run. This would guarantee that the exact code version used in this
158 manuscript will be preserved permanently. [https://docs.github.com/en/repositories/archiving-a-github-
159 repository/referencing-and-citing-content#issuing-a-persistent-identifier-for-your-repository-with-
160 zenodo](https://docs.github.com/en/repositories/archiving-a-github-repository/referencing-and-citing-content#issuing-a-persistent-identifier-for-your-repository-with-zenodo)

161
162 We have made our repository transparent. We have also added it on Zenodo.

163
164 2) Spell-check for the manuscript and code repository (e.g. micorbiome -> microbiome etc.)

165
166 We have fixed the typos.

167
168
169

Re: mSystems01068-24R1 (Designing host-associated microbiomes using the consumer/resource model)

Dear Prof. Purushottam Dixit:

In addition to your other modifications in the current draft of the manuscript, thank you for carefully reviewing your code and ensuring it is reproducible.

Your manuscript has been accepted, and I am forwarding it to the ASM production staff for publication. Your paper will first be checked to make sure all elements meet the technical requirements. ASM staff will contact you if anything needs to be revised before copyediting and production can begin. Otherwise, you will be notified when your proofs are ready to be viewed.

Sincerely,
John Gibbons
Editor
mSystems